# *The Revised Third Cambridge Catalogue* at 60: To Jet or Not to Jet . . .

**Peter Barthel** [1,*] and **Paolo Padovani** [2]

1 Kapteyn Institute, University of Groningen, 9747AD Groningen, The Netherlands
2 European Southern Observatory, D-85748 Garching bei München, Germany; ppadovan@eso.org
* Correspondence: pdb@astro.rug.nl

*The Revised Third Cambridge Catalogue of Radio Sources* (in the northern sky), or 3CR, published sixty years ago by Bennett (1962) [1,2], has been of crucial importance in the study of active galaxies and active galactic nuclei, or AGN, near and far. Few astronomers knew what a radio galaxy was before the publication of the milestone 3CR catalogue. The discovery of the enigmatic quasars goes back to the optical identification and subsequent study of 3CR sources. Generations of astronomers have studied samples of 3CR radio galaxies and quasars and individual 3CR objects. They know the numbers of their 3CR favourites by heart, and some have specialized in the study of one or a few 3CR sources over decades. In particular, 3CR405—Cygnus A—and 3CR274—Virgo A—became the Rosetta Stone radio galaxies. A most useful compilation of the extragalactic 3CR radio sources and their optical properties was published by Spinrad et al. in 1985 [3]. All these 3CR studies shaped our knowledge of extragalactic radio sources: their morphologies with radio lobes, hotspots, compact nuclei, and extended jets; their occurrences; their energy sources; their host galaxies and environments; their interrelations; etc. The desire to understand these black-hole-accretion-driven powerful objects has been the primary driver in the development of radio interferometry from the 1960s to now, including intercontinental very-long-baseline interferometry and culminating in today's worldwide Event Horizon Telescope.

Globally speaking, extragalactic 3CR sources are either high-radio-power, efficiently accreting radio galaxies and quasars with edge-brightened double-radio morphologies, or low-power, inefficiently accreting radio galaxies, with edge-darkened morphologies. The former, mostly of the so-called FR2 class (Fanaroff and Riley, 1974 [4]), have a much smaller space density than the latter (the FR1 class), but much larger radio luminosities, and, thus, can be observed over large distances or look-back times. Radiatively efficient accretion, whereby the black hole mass consumption expressed in energy is a few per cent of the object's maximum radiation (this ratio is defined as the Eddington ratio, where the Eddington power is $1.3 \times 10^{38}$ erg s$^{-1}$ for one solar mass), occurs in massive host galaxies with dust and young stars. In contrast, inefficient accretion, at Eddington ratios of less than about one per cent, occurs in ultra-massive hosts without dust and young stars. The difference relates to the black hole fuel supply, that is to say, the mass to be accreted, and the radio jet formation mechanism. As reviewed, for instance, by Heckman and Best (2014) [5], the properties of the radio source host galaxies, as well as their environments, are important in determining the so-called 'mode of accretion'.

Jets (foremost traced in the radio) can have a profound impact on the surrounding host interstellar medium, influencing star formation rates, gas dynamics, and the overall host galaxy structure. The feedback from AGN jets—both positive and negative—is believed to regulate star formation and control the growth of galaxies, impacting their evolution over cosmic time (Fabian, 2012 [6]). Ultra-massive FR1 host galaxies are generally red and dead as a likely consequence of the jet-driven radio source. On the other hand, massive star-bursting,

i.e., blue-and-alive FR2 hosts are all but unusual, particularly in early epochs (Podigachoski et al., 2015 [7]). Two prototypical examples are shown below, in Figures 1 and 2.

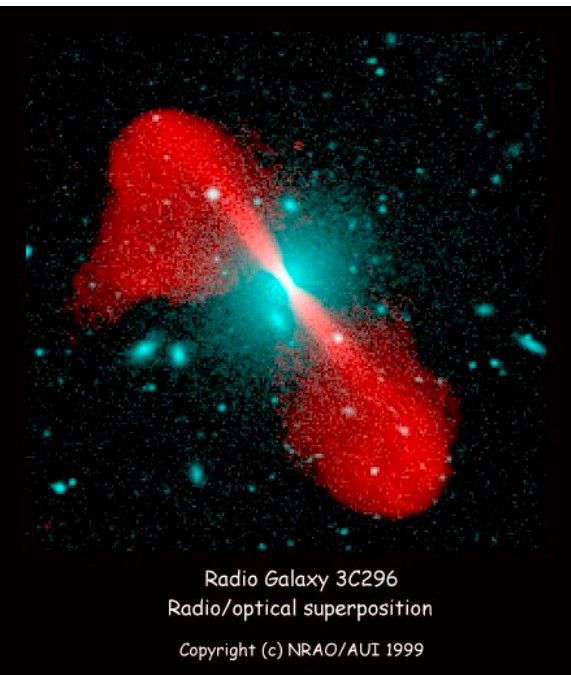

**Figure 1.** Radio-optical images of the prototypical FR1 radio galaxy 3CR296, with prominent jets and diffuse radio lobes, and an ultra-massive host galaxy. Radio emission is shown in red, optical emission in green (NRAO/AU).

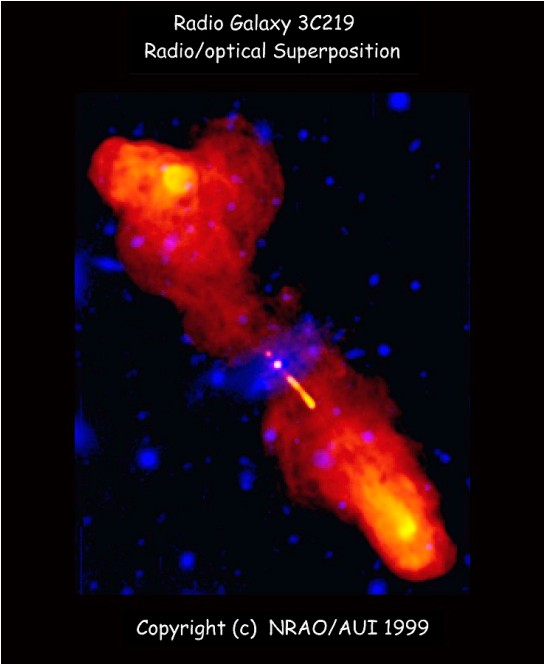

**Figure 2.** Radio-optical images of the prototypical FR2 radio galaxy 3C219 with prominent radio lobes with hot spots at the outer edges, and a massive host galaxy. Radio emission is shown in red, optical emission in blue (NRAO/AU).

Studies on the radio morphologies of 3CR sources have led to orientation-dependent unification schemes. Concerning these, the anisotropic obscuration (toroidal dust configurations surrounding the accretion disk and central black hole) and relativistic effects in the jets conspire to produce seemingly different appearances for the same type of object(s). Anisotropic obscuration is acting in the class of efficiently accreting objects, not in the inefficient class. However, this is not uniformly true in the FR1-FR2 transition radio luminosity regime; certain FR2 radio galaxies with optical spectra of low excitation lack an obscuring torus and a bright UV/X-ray nucleus. In these objects, the active accretion must have (temporarily) halted, implying that the setting up of a torus must be connected to active accretion. All these facts indicate that the host galaxy properties are physically connected to the circumnuclear properties. Doppler beaming in relativistic jets and associated flux-boosting effects are present in both accretion modes, affecting all wave bands from radio to gamma rays. Antonucci (2023) [8] describes the issues in depth.

However, with a detection limit of 9 Jy at the observed frequency of 178 MHz, the six-decade-old 3CR survey detects only a tiny fraction of all AGN: only about three hundred extragalactic radio sources appear in the catalogue. Sensitive radio surveys in the 1970s and 1980s, at a higher frequency and angular resolution, detected many more at $10^{-3}$ to $10^{-4}$ of the 3CR detection limit. The classes of "radio-loud" and "radio-quiet" AGN were proposed. In the 1990s, when detection limits of about $10^{-6}$ of the 3CR limit were reached, it appeared that the population of radio-emitting AGN blended in with the population of radio-emitting star-forming galaxies at very low flux density levels (e.g., Padovani, 2016 [9]). In fact, (faint, diffuse) radio emissions from cosmic-ray electrons interacting with magnetic fields in regions of active star formation provide a reliable and quantitative tracer of galaxy star formation. Radio observations, therefore, complement other wavelength regimes (such as infrared and optical) in providing a comprehensive view of the ongoing star formation activity in galaxies (Condon, 1992 [10]).

Recent studies, reaching a sensitivity seven orders of magnitude higher than 3CR - see Figure 3 for an example - have shown that the decades-old radio-loud versus radio-quiet distinction is unphysical. Every galaxy, be it active or non-active, emits radio waves connected to *stellar* processes. Radio-silent active galaxies do not exist, simply because radio-silent galaxies do not exist. The question is, therefore, are there AGN without black-hole-accretion-driven radio emissions, that is to say, at a totally insignificant level below the radio emissions generated by the host galaxy? The answer, revealed through investigation into the (also notably incomplete) X-ray-selected AGN population is: accretion-driven radio sources comprise about half of the total AGN population (Radcliffe et al., 2021 [11]).

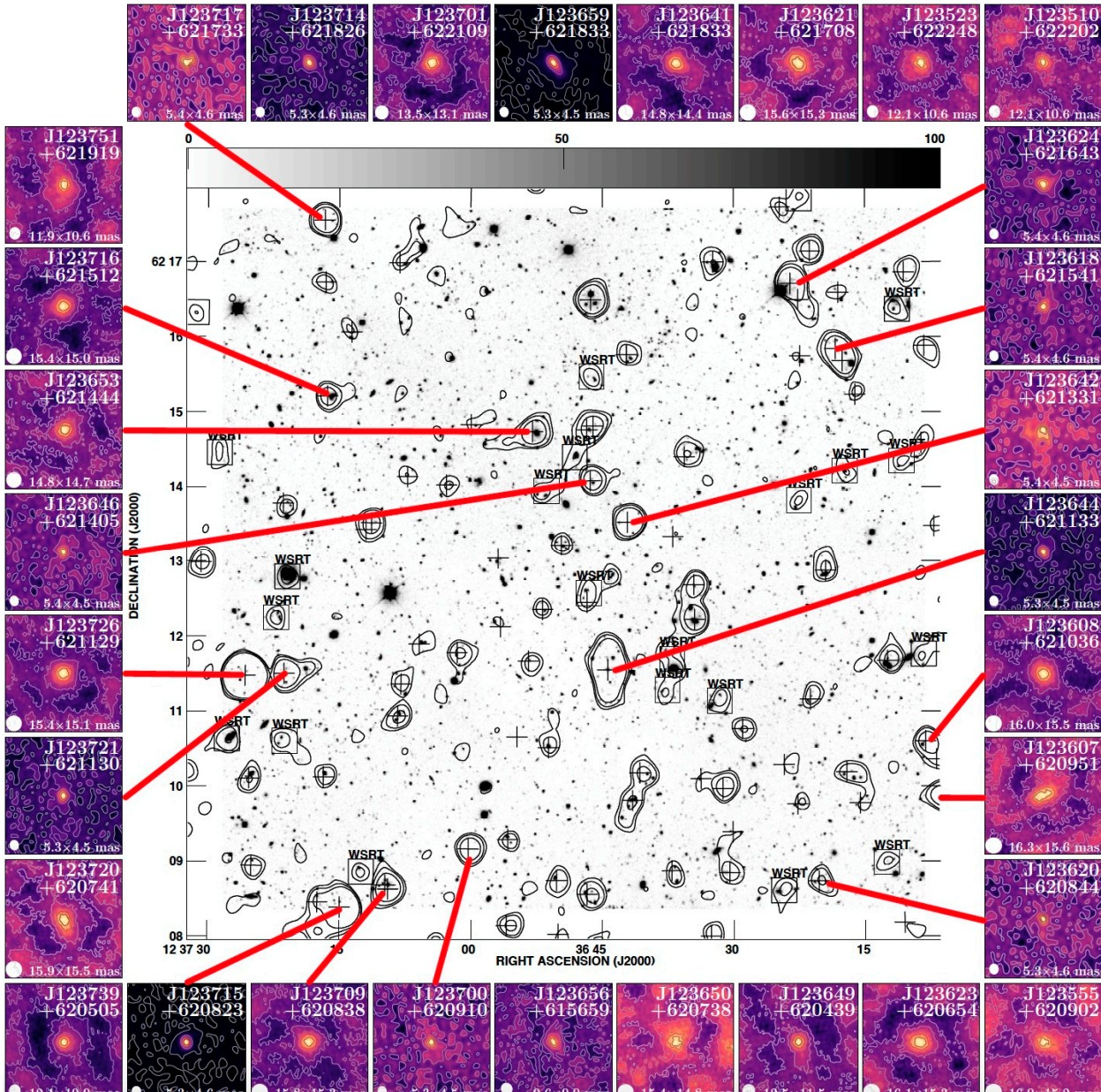

**Figure 3.** The 31 VLBI-detected (i.e., jetted, down to 10 microJy) AGN among the microJy-strength radio source populations in the GOODS-N field. The radio-optical overlay is from Radcliffe et al., 2018 [12], and is reproduced with permission © ESO.

Central black hole accretion occurs in many kinds of galaxies, and the black holes feed themselves in various ways, in intermittent episodes of some ten to hundred million years in duration. In the meantime, those galaxies will form stars at a gradual pace or in short, rapid bursts. Given the established black hole scaling laws, these processes are astrophysically interconnected in a way that is not currently understood. While accreting, galaxy nuclei may or may not develop jets, with associated large-scale radio emissions (Padovani, 2017 [13], Radcliffe et al., 2021 [11]). Regardless of the accretion strength—that is, regardless of the luminosity of the AGN—the nuclear accretion processes may result in a *jetted* (weak, moderate, strong, ultra-strong as in 3CR) or a *non-jetted* AGN.

To conclude, sixty years after the publication of the seminal 3CR catalogue, astronomers are getting to grips with the nature of the radio emissions in active galaxies: black-hole-accretion- and star-formation-driven radio emissions occur in concert, with greatly varying

contributions. However, what exactly drives the formation of jets remains to be solved. To jet or not to jet—that is the question! (December 2023)

Peter Barthel and Paolo Padovani have been actively researching AGN and active galaxies since the mid-1980s, employing radio, infrared, optical, X-ray, and gamma-ray telescopes on the ground and in space.

**Funding:** This research received no external funding.

**Conflicts of Interest:** The authors declare no conflict of interest.

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
