# Peer review of "The Revised Third Cambridge Catalogue at 60: To Jet or Not to Jet …"

_galaxies, doi:10.3390/galaxies12010003_

Round 1

Reviewer 1 Report

Comments and Suggestions for Authors

Dear Editor,

This is a concise and at the same time quite interesting and informative paper on the key catalog of the northern sky extragalactic radio sources representing bright active galactic nuclei of different optical classes, including powerful quasars and nearby radio galaxies. The catalog emerged in parallel with the appearance of the VLBI method and its first successful intercontinental realization, still holding the record on angular resolution ever achieved. The authors discuss crucial questions re. our current understanding of the jetted activity of radio galaxies and tie its wide range of activity level with a phase which a source undergoes.

I do not have any concerns with the paper. The only minor comment from my side is:
- p.2-3: Please remind readers the detection limit of the 3CR, relative to which the corresponding detection limits of later and more sensitive catalogs are mentioned, a
s the absolute values are also important.

The paper merits publication in the journal.

Author Response

Dear referee -

thanks for your positive report. We have added the 3CR detection limit, and given a little more information about the later surveys - at higher frequency and angular resolution. Cf. suggestions by another referee we have also written some lines about the radio source - star formation symbiosis.

Sincerely, Peter Barthel and Paolo Padovani

Reviewer 2 Report

Comments and Suggestions for Authors

Abstract:  It is too short.  It should list a very few main points that are dealt with in the main body of the paper and explain why they are important.

1.       Paper itself:  It, too, is too brief.  A few main points and concepts should be made, and each one should carefully be described and expounded.  To wit:

a.       In 1970 the basic model of AGN was created in which gravitational potential energy is converted to kinetic and radiation energy near a supermassive black hole.  This was generally accepted by 1990.  3CR was instrumental to the development of this model.

b.       Low energy (radio) is concerned mainly with synchrotron radiation, which when Compton boosted, yields Very High Energy (VHE) emission, with energies as high as 20 TeV or more.  Again, here 3CR is critical at establishing the nature of the synchrotron radiation, and multiwavelength observations are very important.  (This explains, perhaps, why optical jets are not to be seen with rare exception.  That waveband is jumped over in the Compton process.)

c.       (Magneto)hydrodynamical numerical model simulations were developed as increasing computational power allowed.  The morphology, initial conditions, and presence of jets are due in large part to 3CR.

2.       Specifics in the paper:

a.       Line 46:  Mention examples of orientation-dependent unification:  Radio galaxies (3CR shows this), BL Lacs, and quasars.

b.       Line 56:  Elaborate on Doppler beaming in relativistic jets.  In particular, we have bulk motion (γ ≈ 10) responsible for superluminal motion and plasmons as seen in radio (VLBI), and (γ ≈ 106) particle motion responsible for Compton boosting.

c.       Line 88:  Consider radio observations of more local AGN, such as the centers of M 31 and the Milky Way.

Author Response

Thank you for your review.
As discussed with the editor, this is not a research paper discussing (jet) research design, methods and results but an ultra-short review - at the occasion of the publication six decades ago of the 3CR survey - of the impact of the 3CR survey on our evolving understanding of the key aspects of extragalactic radio sources.
The revision also touches briefly on the AGN - star formation interplay, and we have made the Abstract and the Key Words more specific.

Reviewer 3 Report

Comments and Suggestions for Authors

This paper provides a brief introduction to the history of the role of the 3CR survey in identifying extragalactic AGN-related radio sources and the evolution of community understanding of the physical meanings of variations in the character of the radio properties and associated nuclear properties of the host galaxy. This is a story worth telling, but, as told, the connections between the radio source story and the galaxy evolution story are weak. This weakness may reflect, in part, allowed space limitations, but I do recommend that before the paper is accepted for publication the links between the two stories be made more explicit and, thus, clearer. The title of the special issue, after all, specifically targets the SYMBIOSIS BETWEEN RADIO SOURCES AND GALAXY EVOLUTION.

For the most part the paper seems well written, although there are a few scattered issues that need need clarification. For example, near the end of page 1 the discussion of radiative efficiency alludes to the "Eddington ratio", but leaves it undefined.

Author Response

Dear referee -

thanks for your positive report. We have - within the scope of this mini-review - written several new lines about the radio source - star formation symbiosis, following up on your entirely correct request. Cf. suggestions by another referee we have also added the 3CR detection limit. 

Reviewer 4 Report

Comments and Suggestions for Authors

The brief perspective article reviewed the impact of 3CR catalog. I have no significant concerns or comments. It would be nice if the abstract can include more scientific points.

Author Response

We have updated the Abstract and the Key Words and are grateful for your positive report.

Round 2

Reviewer 2 Report

Comments and Suggestions for Authors

Much improved.